# Mass Spectrometry to Study Chromatin Compaction

**DOI:** 10.3390/biology9060140

**Published:** 2020-06-26

**Authors:** Stephanie Stransky, Jennifer Aguilan, Jake Lachowicz, Carlos Madrid-Aliste, Edward Nieves, Simone Sidoli

**Affiliations:** 1Department of Biochemistry, Albert Einstein College of Medicine, Bronx, NY 10461, USA; stephanie.stranskylauar@einsteinmed.org (S.S.); jake.lachowicz@einsteinmed.org (J.L.); edward.nieves@einsteinmed.org (E.N.); 2Department of Pathology, Albert Einstein College of Medicine, Bronx, NY 10461, USA; jennifer.aguilan@einsteinmed.org; 3Department of System & Computational Biology, Albert Einstein College of Medicine, Bronx, NY 10461, USA; carlos.madrid-aliste@einsteinmed.org; 4Department of Developmental & Molecular Biology, Albert Einstein College of Medicine, Bronx, NY 10461, USA

**Keywords:** chromatin, DNA methylation, histone, mass spectrometry, post-translational modification, proteome

## Abstract

Chromatin accessibility is a major regulator of gene expression. Histone writers/erasers have a critical role in chromatin compaction, as they “flag” chromatin regions by catalyzing/removing covalent post-translational modifications on histone proteins. Anomalous chromatin decondensation is a common phenomenon in cells experiencing aging and viral infection. Moreover, about 50% of cancers have mutations in enzymes regulating chromatin state. Numerous genomics methods have evolved to characterize chromatin state, but the analysis of (in)accessible chromatin from the protein perspective is not yet in the spotlight. We present an overview of the most used approaches to generate data on chromatin accessibility and then focus on emerging methods that utilize mass spectrometry to quantify the accessibility of histones and the rest of the chromatin bound proteome. Mass spectrometry is currently the method of choice to quantify entire proteomes in an unbiased large-scale manner; accessibility on chromatin of proteins and protein modifications adds an extra quantitative layer to proteomics dataset that assist more informed data-driven hypotheses in chromatin biology. We speculate that this emerging new set of methods will enhance predictive strength on which proteins and histone modifications are critical in gene regulation, and which proteins occupy different chromatin states in health and disease.

## 1. Introduction

Chromatin accessibility has a fundamental role in a wide range of biological processes including gene regulation and DNA repair. While the dogma open chromatin = transcribed genes still stands, there is still much unexplored in understanding which molecular mechanisms regulate chromatin accessibility, which are inherited, and which are failing in disease pathogenesis [1]. Numerous reviews discuss the genomics strategies and the know-how on chromatin accessibility in much detail, including an excellent recent review of Klemm et al. [2]. However, most of the data we have available come from DNA-centric approaches, i.e., high-throughput sequencing. Protein-centric studies are far less common. One exception might appear to be chromatin immunoprecipitation-sequencing (ChIP-seq) [3,4,5], as it maps protein occupancy on the chromatin; but still it provides DNA reads rather than protein data.

Proteins are actually critical players in chromatin state and dynamics. In eukaryotic cells, DNA exists in close association with histone proteins, which form nucleosomes every ≈147 bp of DNA. Canonical histones are H2A, H2B, H3, and H4, and they are bound especially in heterochromatic silenced domains by the linker histone H1 [6]. Histones are highly decorated by post-translational modifications (PTMs), which contribute to chromatin accessibility directly through their charge state or indirectly by recruiting other proteins involved in chromatin modulation (Figure 1). An example of direct accessibility is histone methylation that occurs on lysine and arginine residues. In mammalian cells, one of the best-studied marks is H3K9me2/3 (histone H3 tails di/trimethylated on the lysine residue 9), which are mainly found in constitutive heterochromatin [7]. Another example is histone acetylation, or rather acylation; acyl groups added on lysine residues neutralize the positive charge of the amino acid, which reduces the electrostatic interaction with the negatively charged DNA. For instance, acetylation of histone H3 is associated with active chromatin and plays a fundamental role in transcriptional activation [8]. Beside the abundant acetylation, histone propionylation, malonylation, crotonylation, butyrylation, succinylation, glutarylation, 2-hydroxyisobutyrylation, and β-hydroxybutyrylation are other examples of acylations detected on histone proteins [9]. Proteins with domains that bind those modifications are defined as “readers”. Some of them are transcription factors, whose duty is to recruit RNA polymerases for transcription [10]. Selected readers contribute to chromatin remodeling, i.e., rearranging DNA from a compacted to transcriptionally accessible state, or vice versa. These chromatin remodelers may directly bind DNA motifs rather than histone modifications, e.g., the SWI/SNF complex has high affinity for DNA and it is required for the enhancement of transcription by many transcriptional activators in yeast [11]. As well, other chromatin readers recognize silencing marks and contribute to chromatin compaction. An example of a protein involved in maintaining condensed heterochromatin is the Heterochromatin Protein 1 (HP1), which recognizes and binds H3K9me3 [12]. The spatial positioning of chromatin in the cell nucleus is also contributing to its accessibility [13]. For instance, Lamina-Associated Domains, or LADs, are heterochromatic domains sequestered at the nuclear periphery. Interestingly, those domains are heavily decorated by a selected histone mark, H3K9me2 [14]. As well, the nuclear pore complex associates primarily with DNA regions silenced by the Polycomb Repressive Complex, at least in Drosophila [15]. In summary, the fine-tuning of protein–DNA interactions, protein–protein interactions, and protein modifications (especially histones) are critical contributors to chromatin accessibility. Intuitively, mutations and anomalous protein regulations can have a dramatic effect on the cell phenotype.

Events like UV exposure, smoking, viral infection, cancer, and aging correlate with chromatin decondensation, i.e., large and small heterochromatic domains become euchromatic (accessible and prone to transcription) [16]. In fact, chromatin decondensation is not only an issue in terms of uncontrolled gene expression; chromatin domains decorated by H3K9me2/3 are rich in DNA repetitive units such as transposons, ALUs, and other satellite regions [17], and their accessibility to the transcriptional machinery is harmful for the cell [18]. Together, DNA methylation, histone PTMs, and non-coding regions ensure proper chromatin conformation and promote genome stability [19].

One of the main features of cancer is the (epi)genome instability. The transition from normal tissue to cancer is sometimes characterized by changes in the distribution of H3K9me2/3, in HP1 expression levels [12] and by regional loss of heterochromatin which, in turn, become euchromatin [16]. Additionally, DNA hypomethylation of CpG dinucleotides in the pericentromeric region of the chromosome might be involved in many types of tumors [20]. All these changes directly affect the transcriptional activity and genomic stability, leading to cellular uncontrolled proliferation and metastasis. Interestingly, elevated levels of methylation, especially in gene promoter regions, is related to aberrant silencing of transcription and inactivation of tumor-suppressor genes [20].

Reduced global heterochromatin, altered histone marks, and global hypomethylation of DNA have also been associated with aging. Significant changes in global nuclear architecture during physiological aging, as well as altered gene expression, might be triggered by the loss of heterochromatin domains [21]. This global loss was observed in human old fibroblasts and fibroblasts from Hutchinson–Gilford progeria syndrome (HGPS), indicating that several components are shared between normal aging and accelerated aging syndromes [22]. Senescent cells, during aging, present different levels of histone variants. An example is the loss of canonical histone H3.1 and H3.2 and the increase of the histone variant H3.3, which is incorporated into the genome in a replication-independent manner and plays a key role in chromatin maintenance when cells are no longer dividing [21]. MacroH2A is another histone variant that promotes transcriptional silencing and is abundant in SAHF (senescence associated heterochromatin foci), in addition to being a critical regulator of chromatin dynamics during senescence [21,23]. In fact, chromatin structure is under dynamic changes throughout the entire life span of an organism. Among the histone modifications that are known to affect the longevity process, the most important ones are acetylation and methylation of lysine residues. Increased levels of H4K16ac lead to more open chromatin and, in old yeast cells, it correlates with decreased silencing of reporter genes and shortened lifespan. Conversely, reduced levels of H4K16ac is beneficial for longevity in yeast, due to a more closed global chromatin structure [24].

External events also affect chromatin state. DNA damage accumulates with age, but the process can be accelerated by reactive oxygen species (ROS), exposure to UV irradiation, and alcohol intake. Oxidative stress occurs because of ROS accumulation, affecting chromatin and chromatin modifying-enzymes. In general, it can stimulate global heterochromatin loss and modify histones folding and stability, as well as their PTMs, influencing the expression of genes that are normally in a silenced state [25,26,27]. Besides oxidative stress effects, exposure to ionizing radiation leads to less compact heterochromatin, which adopts a more loose structure [28]. At the same time, there is evidence that radiation induces global compaction of chromatin, indicating a potential mechanism to protect genome integrity [29,30]. Metabolites generated during ethanol metabolism can also impact chromatin structure. Animal experiments demonstrated that excessive alcohol intake modifies the mechanisms regulating chromatin remodeling and gene expression by altering the levels of histone acetylation as well as DNA methylation [31].

In summary, maintenance of chromatin structure is essential for proper cellular function over a lifetime. While we have a large set of data on DNA sequence, conformation, and selected proteins defining accessible vs. inaccessible chromatin, a proteomics-centric view of chromatin is still less exploited. What are all the histone modifications defining euchromatin and heterochromatin? What part of the proteome occupies different chromatin domains in health and disease? In this review, we introduce the most widely adopted methods to investigate chromatin accessibility and then we discuss the methods based on mass spectrometry to characterize the chromatin state where the proteome resides. We conclude with examples of available databases reporting properties of the chromatin proteome.

## 2. Popular Methods to Investigate Chromatin Accessibility

Numerous genomic, biophysical, and microscopy-based methods are now established and used routinely for chromatin state analysis (Table 1). What follows is a brief overview of various current methods and their limitations, to define chromatin state and dynamics.

Genomic methods to study chromatin include but are not limited to the following. A common method for identifying regions of DNA methylation, or condensed DNA, is Methyl-DNA immunoprecipitation sequencing (MeDIP-seq), which utilizes specific antibodies to pulldown methylated DNA followed by sequencing [32]. The same technique can also be modified to determine regions of hydroxymethyl enrichment utilizing methyl-derivatized DNA followed by immunoprecipitation and sequencing [33]. Similar to MeDIP-seq in terms of identifying condensed DNA, Micrococcal Nuclease sequencing (MNase-seq) provides information on nucleosome concentration of DNA regions by MNase digestion, DNA-protein complex purification or, optional immunoprecipitation, and sequencing [34]. Contrary to techniques just discussed, open chromatin can be identified through assay for Transposase-Accessible Chromatin sequencing (ATAC-seq). ATAC-seq utilizes a hyperactive Tn5 transposase that binds and ligates an adapter to DNA [35]. Adapter ligation is followed by DNA transposition and then sequencing, revealing regions of DNA with increased accessibility [35]. Alternatives of ATAC-seq are assays based on restriction enzyme accessibility [36], DNA methylation induced by methyltransferases with low sequence specificity [37] or DNase I hypersensitive sites sequencing (DNase-seq). Specifically DNase-seq, with similar theory to ATAC-seq although, more labor-intensive and has a larger cell requirement [38], utilizes DNase to digest accessible regions of DNA, followed by purification, and sequencing, to indicate areas of increased accessibility [39]. For details about more specific chromatin interactions, Chromatin Immunoprecipitation sequencing (ChIP-seq) yields information about protein-DNA binding. Antibodies specific for the protein of interest are utilized to pull-down DNA-bound protein followed by sequencing of the pulled-down DNA [3]. Antibodies that are utilized for ChIP-seq can be specific for histones and their post-translational modifications and therefore, used to study chromatin. To get the whole picture of chromatin interaction in the genome, Hi-C (genome-wide chromatin conformation capture) involves crosslinking DNA within close proximity, purification, and sequencing, to determine long-range contacts and chromatin arrangement [40].

Contrary to genomic methods, biophysical methods can give a wealth of information about the fine molecular mechanisms that occur within chromatin. Biophysical techniques that have been utilized include optical tweezers [41], which measures the force generated by a laser for chromatin wrapping and unwrapping to study chromatin assembly, histone displacement, and enzyme force. Förster resonance energy transfer (FRET) has also been utilized [42], which can determine properties such as nucleosome organization and chromatin states by monitoring FRET dyes at specific locations.

Although genomic and biophysical methods make up a large portion of the chromatin literature, microscopy techniques can provide information that the others cannot. One such piece of information that microscopy can obtain is through visualizing chromatin domains. One method, stochastic optical reconstruction microscopy (STORM), achieves a resolution of 20–30 nm and has been demonstrated to observe chromatin segregated nanoclusters, dispersed nanodomains, and compact large aggregates, all of which form with different histone modifications [43].

While the techniques explained above generate a great amount of chromatin information, newer and lesser-performed techniques are available to pioneer forward and build upon lacking areas of information. A newer technique, demonstrated with cysteine-lacking histones, includes probing histone surface accessibility [44], in which a single residue is replaced with cysteine in the region of interest and treated with biotin-maleimide. The biotin-maleimide will interact with the cysteine, thus allowing nucleosome purification by a streptavidin-conjugated resin after digestion with micrococcal nuclease (MNase). The resulting DNA can then be sequenced to determine DNA regions of increased histone accessibility.

The techniques explained above can all be informative of chromatin state; however, each have limitations. For example, many sequencing techniques, such as ChIP-seq, can be costly per sample and provide broad information in which combination with more sensitive techniques will be needed to investigate in-depth mechanisms. Hi-C data can be noisy and can lack long-range contacts [45]. Other techniques, like microscopy-based ones, are limited by the type of fluorescent probe and image processing capabilities. Biophysical techniques output a limited and very specific dimension of data. Additionally, tissue specific issues can occur such as autofluorescence of intact tissue in STORM [46], live tissue damage caused by high light intensities in optical tweezers [47,48], and when analyzing frozen tissues with ATAC-seq [49].

Mass spectrometry is an unbiased and multidimensional tool to analyze spatially resolved proteomes, including histone modifications (gatekeepers of chromatin readout) [50]. A number of reviews have described accurately how mass spectrometry has made rapid progresses in accurately identifying and quantifying single and combinatorial histone post-translational modifications [51,52]. However, results are usually interpreted by utilizing previous knowledge on the function of these histone marks, as their accessibility is generated by different methods such as those described above. As well, identification and quantification of the chromatin-associated proteome is now routine analysis, with commercial kits that are available for the rapid enrichment of the chromatin-bound proteins. These methods remain fundamental in addressing critical biological questions; for instance, by isolating the chromatin proteome it is possible to study whether a protein binds the DNA only after a given treatment, or whether a transcription factor associates to the chromatin only when phosphorylated. In this excellent review of Nguyen and colleagues [53], it is discussed how emerging technologies utilizing mass spectrometry can be used to study the nuclear vs. cytoplasmic proteome, including the shuttling of proteins between the two compartments upon stimuli, development, or simply circadian rhythm. In addition, mass spectrometry-based proteomics can be utilized to identify proteins binding to synthetic oligonucleotides to define interactors of specific DNA sequence motifs [54,55]. However, mass spectrometry is still scarcely utilized to generate data that directly assess whether a protein or a DNA element is e.g., in an actively transcribed domain. We will first introduce the use of mass spectrometry to quantify DNA modifications, benchmarking chromatin accessibility, and then focus on applications to directly assess degrees of chromatin accessibility of the proteome.

## 3. Mass Spectrometry to Study Chromatin State: First Steps with Nucleotide Modifications

DNA methylation is a known marker of DNA silencing which regulates gene expression and epigenetics inheritance [56,57]. Chromatin domains with methylated DNA are associated with compacted heterochromatin states. This prompted the need for genome-wide DNA methylation analysis and resulted in the continuous evolution of various analytical methods involving bisulfite reactions, the use of methylation-sensitive restriction enzymes, radiolabeling, immunoassays, methylation specific PCR, microarray technology, next generation sequencing, thin layer chromatography (TLC), and reversed phase high pressure liquid chromatography (RP-HPLC) and with mass spectrometry [58]. However, the two most popular methods remain ELISA and mass spectrometry [59]. Immunoassays are arguably faster and simpler, but they tend to be more variable due to non-specificity issues. Mass spectrometry is considered as the “gold standard” due to the high sensitivity and specificity of the technique, but it is not as robust and straightforward.

Paper, thin layer, ion exchange and gas chromatography coupled to either UV or mass spectrometry are historical methods to separate and quantify the four major DNA bases (G, C, A, T) and the methylated DNA bases (5mdC and 5hmdC). In 1980, Kuo and colleagues successfully used C_18_ chromatography coupled to UV detection to quantify 1–2% 5mdC from DNA of calf thymus and salmon sperm [60]. DNA was previously digested into individual nucleosides using DNase I, nuclease P1, and alkaline phosphatase. Another study reported the use of electrophoretic derivatization and electron-capture negative chemical ionization combined with moving belt liquid chromatography-mass spectrometry to quantify 5mdC and 5hmdC [61]. The method has then evolved including a combination of (1) DNA hydrolysis using HpaII and MspI restriction nucleases, (2) electron ionization gas chromatography, (3) C_18_ chromatography, and (4) hybridization analysis using a series of probes. Together, this was used to map the methylated regions of DNA containing an actively and differentially expressed somatic H1 histone gene from sperm, embryo, and adult tissues of *Chaetopterus* worm [62].

DNA methylation was quantified in disease states like leukemia using urine samples [63]. Chromatographic separation has required optimization, mostly because small molecules like nucleosides have weak hydrophobic interaction with reversed-phase C_18_ columns. Further optimization included varying methanol solvent to decrease surface tension, addition of acetic acid for protonation and found that ammonium acetate/methanol (88:12 *v*/*v*) is the best for both chromatographic separation and detection in mass spectrometry using electrospray ionization [63]. The addition of two different RNAses and re-precipitation of DNA was importantly discussed to at least minimize possible interference from 5-methylcytidine residues from tRNA and rRNA contaminants [64]. Song and colleagues [65] reported a chromatographic separation for the efficient separation and detection of 5mdC and 5hmdC from the other four deoxyribonucleosides and methylated RNA nucleoside contaminants by electrospray ionization tandem mass spectrometry using a triple quadrupole mass spectrometer from genomic DNA. Methylated DNA was also analyzed using mass spectrometry in embryonic stem cells by measuring 5hmdC and 5mdC [66].

Ito and colleagues discovered that 5mC is not only converted to 5hmC but also into 5-formylcytosine (5fC) and 5-carboxylcytosine (5caC) by TET proteins [67]. Those modified nucleosides represented a greater challenge for quantification due to their lower abundance; 1–20 × 10^6^ cytosine for 5fC and 3 × 10^6^ cytosine for 5caC. In fact, they required modifications to the chromatographic setup to achieve sufficient sensitivity [59]. Chemical derivatization using 2-bromo-1-(4-dimethylamino-phenyl)-ethanone (BDAPE) was introduced to selectively label 5mdC, 5hmdC, 5fdC, and 5cadC [68]. This enhanced sensitivity of 35–123-fold compared to un-derivatized cytosine modifications [59,68]. More recently, hydrophilic interaction liquid chromatography (HILIC) [59,69] and porous graphitic carbon (PGC) [70] have emerged as potential alternatives to C_18_-based chromatography for nucleosides. However, a robust platform for nucleoside quantification using mass spectrometry is still less common in research labs than one could expect. In summary, quantification of DNA modifications has been the pioneer of chromatin state analysis using mass spectrometry since its establishment about 40 years ago. However, for a comprehensive overview of molecular components of accessible and inaccessible chromatin, a protein perspective is required.

## 4. Multi-Dimensional Histone Modification Analysis Using Mass Spectrometry

Mass spectrometry is currently the only technique that can identify and quantify in a large-scale manner the relative abundance of histone PTMs [71]. Other techniques, such as Western blotting, ELISA, or immunohistochemistry, may be used for histone PTM quantification. However, all of these methods rely on specific antibodies, which may not be readily available for some PTMs or may provide inaccurate quantitation due to cross-reactivity or epitope masking [72,73]. Mass spectrometry is used to quantify both single and combinatorial histone PTMs [74], although it is important to note that extracted histones are decoupled with their original DNA location and thus this analysis does not allow to define the genome-wide distribution of histone marks. Creative approaches have been exploited in mass spectrometry to look beyond the sole abundance of histone PTMs; e.g., the group of Anja Groth used metabolic labeling in cell culture to monitor whether newly synthesized or recycled histones were transferred into the newly replicated DNA [75]. Nevertheless, this approach cannot define the location of these histones nor provide direct information about their accessibility.

The traditional peptide centric analysis of histone proteins (bottom-up) is similar to the typical proteomics pipeline. Specifically, proteins are usually digested with the protease trypsin (cleaves after lysine and arginine residues) into short (4–20 aa) peptides for analysis, as both chromatographic separation and detection by mass spectrometry are more robust and sensitive than intact protein analysis (top-down). However, histones are very basic proteins, i.e., rich in arginine and lysine residues, and thus trypsin digestion would result in excessively short peptides for reconstructing their position on the protein. For this reason, derivatization of lysine residues is frequently applied to modify the side chain of lysine residues so that trypsin can only cleave arginine residues and generate proper size peptides [71]. Since 2004, the sample preparation strategy has been periodically optimized and different laboratories have applied different derivatization methods to generate proper size peptides, i.e., from the use of D_6_-acetic anhydride [76], to propionic anhydride [77] to NHS-propionate [78] to phenyl isocyanate [8]. Independently from the protocol applied, it is now clear that by performing chemical labeling of lysine residues assists more confident and accurate PTM quantification. An overview of the different sample preparation strategies, including their advantages and disadvantages, is described elsewhere [79,80,81,82,83].

While histone PTM quantification per se does not provide direct information about chromatin state, the relative abundance of selected histone marks is used to define how accessible chromatin is overall. For instance, hyperacetylation of histone H4 on the residues K5/K8/K12/K16 (in particular K16 [84]) reveals that chromatin is relatively unfolded. As well, the increase in abundance of selected silencing marks has been interpreted as “restricted” chromatin environment, e.g., in schizophrenia [85]. Those are indirect conclusions to the chromatin state, and we should not forget that hundreds of histone marks have been identified but still not assigned to either accessible or inaccessible chromatin. The advent of “middle-down” mass spectrometry showed an even more complicated picture; this strategy is named as such as it is a compromise between the peptide-based approach (bottom-up) and the analysis of intact undigested proteins (top-down). Middle-down utilizes proteases that cleave rare amino acid residues on histone sequences, i.e., aspartic and glutamic acid, to generate intact histone N-terminal tails (50–60 aa) [50]. Identifying and quantifying these long polypeptides is equivalent to mapping co-existing PTMs on the same histone protein, i.e., this approach can be used to define combinatorial PTM codes [50]. Notably, other strategies not based on mass spectrometry have been implemented to study combinatorial modifications; Shema and co-workers developed an antibody-based imaging platform to map co-existing modifications on nucleosomes [86], while Sadeh and colleagues established a method named Combinatorial-iChIP to map genome-wide the co-occurrence of two histone PTMs instead of the typical single PTM analysis of canonical ChIP-seq [87]. These methods offer undeniable advantages; on the other hand, middle-down mass spectrometry is independent from antibodies and it is not limited in the number of co-existing modifications to quantify on a single polypeptide. From middle-down data, it became rapidly clear that histones are very rarely decorated with one or two modifications in the cells, but they rather have 5–8 co-existing marks on the same histone protein [88]. Frequently, those PTMs have unknown biological function or presumed opposite roles on chromatin. Why do they co-exist then? This is still an unanswered question, as there is currently no technology that can define the accessibility on chromatin of hypermodified histone codes.

The differential turnover of nucleosomes has been described in multiple publications [89,90,91], firmly establishing that nucleosomes are exchanged from chromatin multiple times within a cell cycle. This opens an opportunity for mass spectrometry, as protein turnover can be quantified by metabolic labeling (e.g., Zee et al. [92]). In a recent work, we have assessed that metabolic labeling of histones can be utilized to define whether a certain modification is on actively transcribed chromatin or inaccessible [93]. The principle is based on cell cultures feeding on stable isotope labeled amino acids, which are partially incorporated in the histone amino acid sequence (Figure 2). Those histones with higher recycling rates will have a relatively higher heavy/light ratio, and this recycling rate is more frequent on chromatin domains with active transcription. Interestingly, this labeling has the potential to be utilized for middle-down [94], paving the way to the determination of the accessibility on chromatin of combinatorial histone codes. However, part of the current challenge is developing the proper bioinformatics to discriminate signals corresponding to combinatorial modifications vs. partial metabolic labeling.

## 5. Quantifying the Chromatin-State Dependent Proteome with Mass Spectrometry

Proteomics has become a discipline with many applications, most of them contingent with appropriate sample preparation. The routine procedure of sample preparation for proteomics is arguably one of the simplest among the -omics; most extracted or purified proteins are soluble in water, and they can be prepared for mass spectrometry with a rapid three steps procedure, i.e., reduction, alkylation, and digestion into peptides. For this reason, a myriad of alternative procedures have been engineered to enhance sensitivity, specificity, and quantitative dimensions (time, localization, interactions, turnover rate). In other words, we can analyze the proteome of a specific chromatin state if we establish a dedicated chromatin fractionation procedure that allows to physically isolate chromatin fractions and analyze them separately by mass spectrometry or selectively label those domains (Figure 3). In 2003, Shiio and colleagues, in a pioneering study, developed a method to identify and quantify chromatin-associated regulatory factors by a combination of chromatin isolation and mass spectrometry analysis [95]. A recent approach was named “gradient-seq”, a method in which chromatin is cross-linked and afterwards fractionated over a sucrose gradient based on its resistance to sonication [96]. Cross-linked heterochromatic domains generate larger macromolecular structures, which can be separated by smaller accessible euchromatic domains. However, this method is unable to define the histone PTMs associated with more subtle differences in chromatin compaction since it is largely limited to resolving heterochromatin from euchromatin. Hybridization capture of chromatin-associated proteins for proteomics (HyCCAPP) is an approach developed to identify the protein components of alphoid chromatin, which is rich in a highly repetitive class of DNA [97]. Using this method coupled to mass spectrometry, Buxton and colleagues were able to analyze human protein–alpha satellite interactions. Moreover, locus specific proteomics was performed by exploiting the pull-down of a specific DNA region; two protocols named proteomics of isolated chromatin segments (PICh) [98] and insertional chromatin immunoprecipitation (iChIP) [99] were optimized for the direct identification of the bound proteome. PICh is based on nucleic acid probes, while iChIP utilizes antibodies to precipitate specific proteins benchmarking the locus of interest, e.g., CTCF was targeted to identify insulator complexes, which function as boundaries of chromatin domains. Alternatively, synthetic chromatin was also engineered with histone PTMs using ligation to purify proteins binding to selected histone marks [100].

Another option is salt fractionation of chromatin, which can isolate chromatin fractions with completely different genome-wide profiles. According to Henikoff and colleagues, low-salt soluble fraction corresponds to a highly accessible chromatin and high-salt soluble fraction represents the condensed chromatin. The method was validated by showing that the insoluble fraction is enriched in transcriptionally active chromatin and is closely linked to engaged RNA polymerase [101,102]. Extraction and processing using this methodology leads to a complete recovery of chromatin and allows histone epitopes to be readily profiled. The lab of Dr. Henikoff has been pioneering and optimizing this approach, and it has been applied by numerous laboratories to investigate chromatin state in health and disease, e.g., in cancer cells [103] or during viral infection [104]. Indeed, viral infection caused by adenovirus, herpes simplex virus, and Epstein–Barr virus may alter the appearance of the host chromatin. To address this hypothesis, Herrmann and colleagues developed a protocol to fractionate nuclei using a gradient of salt concentration, followed by identification of proteins by mass spectrometry [105].

Several chromatin readers have already been identified, and they have been categorized depending on protein domains that have been characterized to bind different types of histone PTMs. Bromodomains are known to recognize acetylated lysine residues, while PhD fingers can bind to acetylations and also methylations related to active gene transcription such as H3K4me3. Chromodomains are typical of proteins binding histone methylations associated with chromatin silencing. WD40 repeats recognize mainly mono- and dimethylations. Tudor domains are typical of methylated residues such as H3K4me3, H3R17me2, H4R3me2, and H4K20me3. 14-3-3 domains bind serine phosphorylations. MBT and YEATS domains are currently not as characterized, but they are known to bind methylated lysine residues and acyl-modifications such as lysine crotonylation, respectively. Yun and colleagues describe all these domains in a comprehensive review [106].

A strategy combining chromatin immunoprecipitation (ChIP) of histone modifications and mass spectrometry, occasionally named ChroP (Chromatin Proteomics), has been used to analyze protein components characterizing distinct chromatin regions [107,108]. ChroP analysis allows the identification of PTM association between the different core histones within the same mono-nucleosome and indicates variants and readers at functionally distinct chromatin domains. This approach was recently revamped by Iglesias and colleagues [109], who used the HP1 yeast homolog Swi6 to enrich heterochromatic domains in *Schizosaccharomyces pombe*. Similarly, Zukowski and colleagues assembled a repressive heterochromatin domain from purified components, conjugated to magnetic beads, to recruit factors that play a role in regulating heterochromatin function [110].

The requirement of specific antibodies directed against each regulatory protein in ChIP methodology is one of the bottlenecks to studying gene-regulatory factors. To address this issue, catalytically dead RNA-guided nuclease Cas9 (dCas9) has been used to isolate specific genomic regions and their associated proteins. A dCas9-targeted chromatin-based purification strategy was recently used by Tsui and colleagues to identify chromatin-associated proteins and its potential role on the modulation of histone expression [111]. As an alternative strategy, dCas9 has been used in association with the engineering peroxidase APEX2 to biotinylate and identify proteins at a defined genomic loci [112]. According to Myers and colleagues, this method can be used as a complementary approach to ChIP to study proteins related to gene expression and chromatin structure. In a recent approach, using a CRISPR-biotinylation system to tag defined loci, Escobar and colleagues demonstrated that repressed chromatin domains are preserved through local re-deposition of parental nucleosomes during DNA replication. In contrast, nucleosomes decorating active euchromatin do not present such preservation [113]. Active euchromatin regions can also be differentiated from heterochromatin by increasing susceptibility to endonucleases. In this regard, DNase I and MNase endonucleases have been used to explore chromatin bound proteins, which are involved in chromatin structural maintenance [114]. The method described by Dutta and colleagues, based on the differential release of proteins from chromatin upon DNase I and MNase digestions followed by a LC-MS/MS proteomic approach, showed that most of the transcriptional regulation proteins identified were greatly solubilized by both enzymes.

A potential new direction of proteomics for chromatin state analysis could be the selective labeling of proteins accessible to solution by using existing methods for single protein characterization. In mass spectrometry, protein folding can be characterized using hydrogen-deuterium exchange (H/D-X) [115] or Fast Photochemical Oxidation of Proteins (FPOP) [116]. A promising new direction could be to use those labeling techniques in a broader perspective to assess which proteins are accessible on chromatin. More in general, mass spectrometry holds a great potential in determining protein localization via covalent labeling, assuming that this labeling can be easily and selectively incorporated on proteins residing on euchromatic domains.

## 6. Bioinformatics Resources to Study the Chromatin Bound Proteome

Bioinformatics is a vibrant sub-discipline in proteomics. Hundreds of software are available to identify spectra (e.g., MaxQuant, [117]), accurately quantify peptides/proteins (e.g., Skyline, [118]), predict novel unknown PTMs (e.g., MSFragger, [119]), identify new protein complexes (e.g., CoExpresso, [120]), or predict enzymatic activity based on quantified PTM data (e.g., iGPS for kinases, [121]). The canonical workflow is based on matching observed masses from MS and MS/MS spectra with peptides in silico generated from libraries of protein sequences or previously identified spectra, i.e., spectral library search. This is what is normally intended as “database search”. Once protein identification is completed, there are also databases containing data assisting the biological interpretation of results. Specifically for chromatin, there are libraries of protein domains that recognize active and silencing histone modifications [106], and there are software tools that computationally predict these domains. In this final section of the review, we provide an overview of available databases that can be used to generate hypotheses or validate results obtained from proteomics experiment investigating chromatin state (Figure 4). This includes software tools for quantification of histone peptides, as well as to assess chromatin binding domains, databases including repository of known proteins binding to chromatin, and databases including functional data of biological roles of histone modifications.

As described above, mass spectrometry has been used as the method of choice to study chromatin-state as well as for histone PTMs analysis. In order to accurately quantify peptides detected by the LC-MS/MS approach, different software tools were developed. For instance, EpiProfile 2.0 is software purposely designed to extract (un)modified histone peptides in LC-MS/MS runs [122]. The software uses a retention time prediction to enable accurate peak detection. In the same line, Fishtones software was developed to extract ion chromatograms and calculate peptide peaks from histone H3 [8]. Another tool developed on purpose to quantify hypermodified histone peptides is PILOT_PTM [123]. However, these software hardly become widely adopted due to the advanced knowledge still required to acquire and interpret these types of results. Databases that provide details about human histone proteins are also available. An example is HIstome (The Histone Infobase), a manually curated database that contains information of histones, their variants, and the association of specific histone variants or histone modifications with human diseases [124]. This free web-based resource encompasses data of 55 histone variants, 106 distinct sites of their modifications, 152 modifying enzymes along with their biological significance and can be used to understand the role of histone modifications and the chromatin-modifying machinery onto gene activity. Information regarding histones variants, PTMs, and modifying enzymes are categorized into groups for visualization. Since information about “readers” is missing, histone modifying enzymes group is divided only into “writers” and “erasers”. A database including writers, erasers and readers of histone acetylation and methylation (WERAM) for 148 eukaryotic species, including *Homo sapiens*, is also freely available [125]. WERAM provides information about 20,033 non-redundant histone-regulators, including 1337 histone acetyltransferases (HATs), 2504 histone deacetylases (HDACs), 3901 acetyl-readers, 4409 histone methyltransferases (HMTs), 1610 histone demethylases (HDMs), and 10,949 methyl-readers. Data provided by this integrative database is helpful for understanding the molecular mechanisms and regulatory roles of the histone code. ChromDB (*The Chromatin Database*) is another database containing information about chromatin-related proteins that are conserved across eukaryotic species [126]. This tool comprises more than 7000 proteins representing over 30 organisms, including plant genes predicted to encode proteins associated with chromatin remodeling, in addition to animal and fungal proteins to facilitate a comparative analysis of the chromatin proteome. ChromDB sequences are divided into two groups: genomic-based, limited to plant genomes, and transcript-based, including animal and fungal organisms such as *Homo sapiens* and *Saccharomyces cerevisiae*. Different from ChromDB and HIstome databases, DbHiMo platform offers a broad archive and analysis tool for histone-modifying enzymes encoding genes, with emphasis in the fungal species (other than yeast) [127]. This database contains 11,576 histone-modifying enzymes identified from 603 proteomes including 483 fungal, 32 plants, and 53 metazoan species.

Different from the databases described above, HistoneDB2.0 and MS_HistoneDB are databases based on histones sequences. HistoneDB2.0 with variants is a manually curated database that comprehends canonical histones and histone variants, their sequence, structural, and functional features [128]. It allows to organize histones by variant, to provide reference alignments for each variant, to offer curated annotations of variant species features, to find likely orthologs of variants in other species, among other functionalities. MS_HistoneDB, in turn, is dedicated to the study of mouse and human histones sequences [129]. The main advantages are the manually curated databases that contain a non-redundant list of 83 mouse and 85 human histones (canonical and variants), whose nomenclature were unified with the HistoneDB2.0 with variants resource, in addition to having a format that can be directly read by programs used for mass spectrometry data interpretation.

Analysis of proteomics data can be performed with a chromatin binding perspective by loading the regulated subproteome identified into software tools that predict or match specific chromatin-binding domains. In general, specific protein domains can provide insights into protein specific functions. There is a lot known about proteins with domains that recognize either chromatin modifications or DNA motifs and there are software tools that computationally predict these domains. SMART (Simple Modular Architecture Research Tool), Pfam, and Prosite databases are resources for identification and annotation of complete protein domains and the analysis of domain structure [130,131,132]. Among others, these databases contain domains found in signaling, extracellular and chromatin-associated proteins. While Pfam database aims to cover as much of proteins sequences as possible with the fewest number of protein models, Prosite concentrates on precise functional characterization, which can be used for protein database annotation. Indeed, Pfam contains models for 11,177 protein domains, a higher number compared to 1302 domains found in SMART. However, SMART database has a manual annotation pipeline which leads to partially different protein annotation.

The focus of databases is to display sets of highly curated data and make this information available to the research community. However, one of the challenges relies on the fact that databases require constant updates based on new experimental data and it is not always feasible to keep them curated. Although they contain a large amount of useful data, ChromDB was last updated in February 2015, HIstome in October 2017, and DbHiMo in 2015.

## 7. Conclusive Remarks

Chromatin biology has been immensely enriched by the development of new technologies to analyze chromatin state. Those methods, spanning from ATAC-seq to super resolution microscopy, have allowed to observe new quantitative dimensions of chromatin in health and disease. Mass spectrometry has yet to emerge as routine method to quantify chromatin state and define the composition of (in)accessible chromatin domains. However, it is of paramount importance to understand which proteins occupy specific chromatin domains in health and disease. Each protein has specific functions in the cell depending on its localization and its interactions, and this is intuitively true also for chromatin occupancy. Therefore, protein analysis needs to become a regular analysis when investigating chromatin regulation. In this review, we highlighted the approaches that are currently emerging that combine proteomics with state-specific chromatin analysis, aiming to increase awareness about the importance of analyzing DNA from the protein perspective.

## Figures and Tables

**Figure 1 biology-09-00140-f001:**
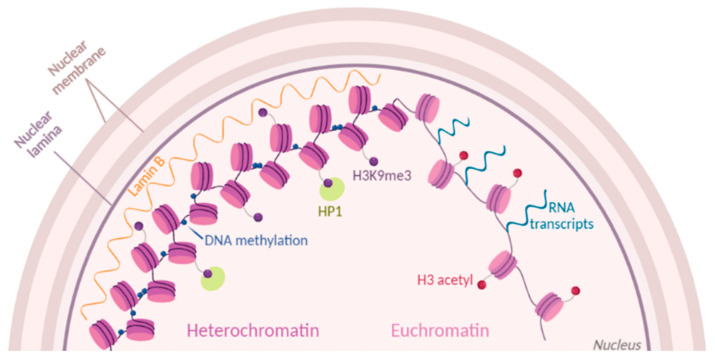
Chromatin dynamics. Chromatin state is modulated by histone post-translational modifications (PTMs) (purple and red dots). Part of condensed heterochromatin is located at the nuclear periphery and is enriched in methylations of histone H3 at the residue K9 (H3K9me3). Readers of this modifications like Lamin B and HP1 maintain the chromatin in a compacted state. Euchromatin is shown as open and accessible chromatin, enriched for histone acetylation (H3 acetyl) and prone to transcription.

**Figure 2 biology-09-00140-f002:**
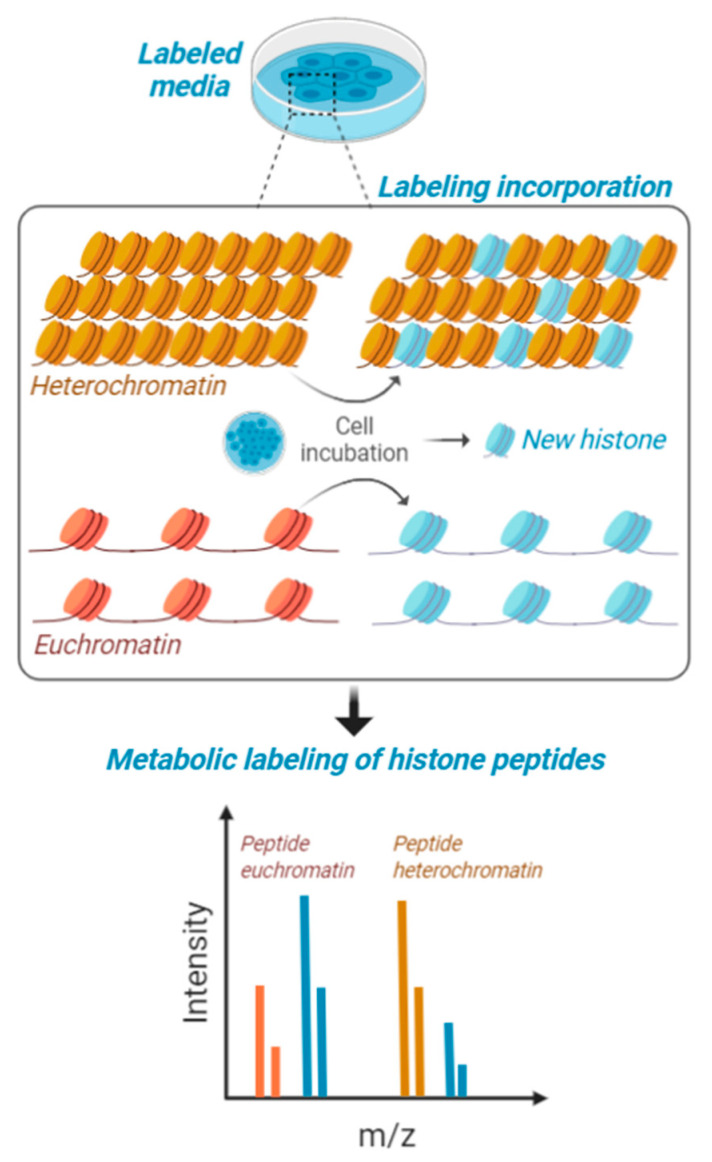
Metabolic labeling of histones peptides. Cells in culture are fed with media containing stable isotope labeled amino acids and are maintained for a certain interval of time to produce about 50% of newly synthesized histones. This interval of time is contingent to their proliferation rate, as the population needs to undergo at least one cell cycle for proper labeling. Heavy amino acids are incorporated in the histone amino acid sequence and then cells are processed for histone PTM analysis. Accessible chromatin (euchromatin—in orange) is labeled with higher rate compared to condensed heterochromatin (in yellow). Isotopic labeling is represented in blue.

**Figure 3 biology-09-00140-f003:**
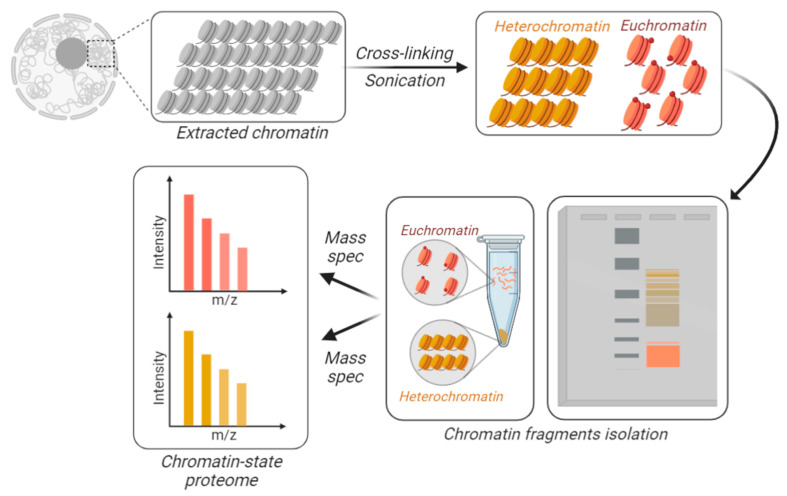
Chromatin-state proteome analysis. To differentially identify and quantify proteins from accessible vs. inaccessible chromatin, the chromatin is physically fractionated into separate tubes. DNA is cross-linked and the extracted chromatin is fractionated based on its resistance to sonication. Larger macromolecules are the result of sonicated heterochromatin (in yellow), while accessible chromatin is sheared into smaller fractions, i.e., euchromatin (in orange). Those fractions can be separated using gels or centrifugation.

**Figure 4 biology-09-00140-f004:**
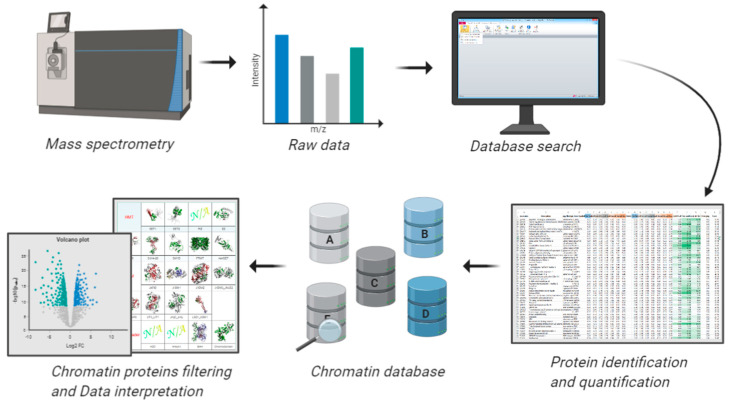
Workflow for analyzing proteomic data using available databases. A common challenge in proteomics is the data analysis aspect, especially because every sample contains background contaminants. A long list of regulated proteins can be overwhelming for biological interpretation, especially for biologists not familiar with -omics datasets. Using databases containing pre-compiled information about the functional role of proteins is frequently helpful to orient around a complex protein list. The databases described in this section offer a platform to filter chromatin-related proteins in proteomics datasets and assist the interpretation of their regulations based on annotated functions.

**Table 1 biology-09-00140-t001:** List of various current methods used to study chromatin state and the type of data collected, chromatin information obtained, difficulty, popularity, and limitations. Difficulty takes into account skillfulness of personnel and instrumentation needed to utilize the technique. Popularity was determined by the number of PubMed method-specific search results on on 24 and 25 March 2020 (<100 = Low, >100 = Medium, >1000 = High, >10,000 = Very High).

Method Name	Type of Data Collected	Information Obtained	Difficulty	Popularity	Limitations
**Stochastic Optical Reconstruction Microscopy (STORM)**	Super-resolution images	Higher-order chromatin structures	High	Medium	Number of fluorescent probes, tissue autofluorescence
**Förster Resonance Energy Transfer (FRET)**	Fluorophore distance, images, and intensity	Nucleosome organization, effector binding, chromatin state	Medium	Very High	Number of fluorescent probes, specific dimension of data
**Optical tweezers**	Force	Chromatin assembly, histone displacement, enzyme force	High	High	Very specific dimension of data, live tissue damage
**Methyl-DNA Immunoprecipitation Sequencing (MeDIP-seq)**	Regions of enriched methylation	Methylated DNA regions	Easy	Medium	Cost, broad information, non-specific binding can occur
**Assay for Transposase-Accessible Chromatin Sequencing (ATAC-seq)**	Regions of accessible chromatin	Exposed DNA regions	Easy	High	Cost, broad information, cryopreserved tissue may not work
**Chromatin Immunoprecipitation Sequencing (ChIP-seq)**	DNA associated with specific proteins	Protein–DNA interactions, histone organization	Easy	High	Cost, non-specific binding can occur
**Micrococcal Nuclease Sequencing (MNase-seq)**	Nucleosome concentration	Condensed DNA regions	Easy	Medium	Cost, broad information
**DNase I Hypersensitive Sites Sequencing (DNase-seq)**	Regions of accessible DNA	Exposed DNA regions	Easy	Medium	Cost, broad information, more time-intensive than ATAC-seq
**Histone surface accessibility**	Regions of accessible histones	Chromatin structure and dynamics	Medium	Low	Requires cysteine-lacking histones
**Hi-C**	Crosslinked regions of DNA	Chromatin arrangement, organization, and long-range interactions	High	High	Broad information, possible lack of long-range contacts

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
