# Peer review of "Mass Spectrometry to Study Chromatin Compaction"

_biology, 2020, doi:10.3390/biology9060140_

Round 1
Reviewer 1 Report
Mass spectrometry to study chromatin compaction
The above review by Stransky et al. tried to convince the reader that Mass Spectrometry (MS) has a greater potential to study chromatin biology. It is generally well written and an interesting review overall. I generally like the figures (please see specific comments on figure below) and the table and the manuscript overall. Fine-tuning the structure and synthesis of expert opinion rather than just summarising facts under certain sections would improve the manuscript as explained below.
- The concept that the authors tried to promote is not novel and has been proposed in previously two well written reviews that the authors failed to refer (Özlem Önder et al., 2015, DOI: 10.1586/14789450.2015.1084231; K.A. Janssen et al., 2017 DOI: 10.1016/bs.mie.2016.10.021). I would like the authors to clearly discuss in their current manuscript that in which ways the current review is different from those two previously published reviews.
- We know that many readers of reviews are general readers and not subject specialists. Readers use reviews as their starting point before diving into specific literature. Therefore, reviews need to be well structured and less confusing. I generally like the structure of the current manuscript but restructuring bit more on the following aspects would make it much clearer and more readable for the general reader.
- It should be clearly stated at the beginning that what aspects of chromatin biology are studied or could be studied with MS in brief (make the biology crisp and clear). These could be further elaborated later under different sections when you discuss different MS based techniques (also see 1c below).
- The basic (general) methodology of MS in chromatin biology is not properly introduced. Figure 4 tried to do that but not described well. “Data base search” vs. “Chromatin data base” in figure 4 would be confusing for the general reader since there is no mention on what are those databases and how data base searches are done during MS data processing (also see Comment 6a).
- Introduction: Lines 75-132 has lots of repetition on changes in chromatin under ageing and cancer and genotoxic stressors etc. Synthesis by the authors is hardly visible here. The section needs to be summarised and structured to improve readability.
- Section “Popular methods to investigate chromatin accessibility”:
- Needs more summarising and restructuring rather than make it appear like a collection of individual sentences unconnected in a paragraph (Lines 146-167).
- ChIP-seq in terms of chromatin biology is not clear in lines 154 – 155. Since ChIP-seq is generally used to study transcription factor binding it should be clearly mentioned as to how ChIP-seq is used to study chromatin. This is also a good place to discuss the limitations and challenges of ChIP-seq to study PTMs etc (rather than line 344-345). How the ChIP-seq is depicted in Table 1 may be enough if a “limitations” column is introduced as mentioned in Comment 4 below
- Table 1 (very nice summary) could be improved as follows.
- Table 1 on its own is sufficient enough to summarise the methods (Section 2.) if you include one more column “Limitations” with few lines of introduction to section 2.
- Table1 “LABS THAT USE IT”: The number of “labs that use it” and “popularity” are the same measure unless many of the publications came from the same very few labs. I would suggest using an integrated measure "popularity index" which combines both these indices.
- Figure 1: How about other methylation marks such as H3K4me3, H3K36me3 etc? Do the authors have a reason to omit them? Should be clear in the figure caption or in the introduction. Authors mention briefly about them later deep in the text. E.g. Lines 267-269. Better to be somewhere in the introduction but could be reiterated again when necessary further down.
- Section 6, Bioinformatic Resources:
- The section should include brief discussion on databases / platforms / resources available for MS data processing (e.g. EpiProfile2.0, https://github.com/zfyuan/EpiProfile2.0_Family) in line with what I proposed in comment 2b.
- The Chromatin databases (DBs) discussed and summarised by the authors need to include how up to date or frequently updated these DBs are so that the readers will be aware how obsolete the data source is. (e.g. as of 12th June 2020, HIstome: not updated since October 2017; ChromDB: Last updated in February 2015 with many organism databases last updated in May 2011; DbHiMo: No updates since 2015).
- Moreover, many databases which are not up to date may contain useful information and there may not be comprehensive alternative sources. However, it is the duty of the expert reviewer to tell the reader that if it is still useful to use them and associated caveats of using such databases (if any). Additionally, authors as experts could point out whether one resource is a better alternative to another etc. This sort of analyses is lacking in the review and should be included in section 6.
- Line 79 “genomic instability”: confusing and needs rephrasing to reflect the real meaning of "genomic instability". Did authors tried to mean "genomic instability commonly refers to” high frequency of mutations and reduced DNA repair?
- Lines 84-85 “Cancer is related to an unstable (epi)genome and it can be preceded by conformational changes of chromatin.”: Confusing and needs rephrasing.
- Line 342 “HP1 yeast homology Swi6”: I suppose authors meant “homolog”?
- Line 345 “…one of the obstacles to studying gene-regulatory factors”: I suppose “bottlenecks to study” would better suit.
Reviewer 2 Report
The authors give a nice overview of methods available to assess chromatin accessibility. In this frame, they focus on a more protein-centric view of this topic, with specific insights in the use of mass spectrometry as a method to study histone proteins and more in general chromatin-bound proteins.
The paper is well written and easy to follow. The content is well explained and deepened. The topic is of great interest and it’s worth to be published.
I only have some suggestions for the authors and some very minor revisions:
- I would reduce the number of keywords.
- Line 50: I would change “catalyzed” with “added”.
- Line 110: Replace “play” with “plays”.
- Line 124: Replace “Reactive oxygen species” with ROS.
- Line 129: Double space after dot.
- Line 165: Replace “Forster” with “Förster”. Also in Table 1.
- Line 175: Replace “MNase” with “micrococcal nuclease (MNase)”.
- Table 1: I would remove the “LABS THAT USE IT” column. Even though this information is different from that included in the “POPULARITY” column, in my opinion retaining both columns is quite redundant. Indeed, the two corresponds in majority of the lines. I think “POPULARITY” is more informative than the other.
- Section 3: I found it very interesting but a little bit too much detailed bearing in mind the topic of this review and if compared to the following sections. I would suggest the authors to shorten this paragraph.
- Line 201: “performance” should be replaced by “pressure”.
- Line 208: Replace “et al.” with “and colleagues”.
- Line 224: I would add a reference.
- Line 227: Replace “et al.” with “and colleagues”.
- Line 234: Replace “et al.” with “and colleagues”.
- Section 5: I would suggest the authors to add in this section some more examples of applications taken from recent research articles, g. linked to viral infection and/or cancer in chromatin state remodelling.
- Line 321: Replace “fraction” with “fractions”.
- Line 324: Replace “et al.” with “and colleagues”.
- Line 326: Double space after dot.
- Line 333: Replace “et al.” with “and colleagues”.
- Line 337: Replace “et al.” with “and colleagues”.
- Line 342: Replace “et al.” with “and colleagues”.
- Line 348: Replace “et al.” with “and colleagues”.
- Line 351: Replace “et al.” with “and colleagues”.
- Uniform “prosite” writing between lines 424 and 428.
Reviewer 3 Report
The review manuscript by Stransky et al. provides an interesting protein-centric, MS-based, view of the field of chromatin biology. The authors provide a broad, timely review of emerging techniques to study epigenetics, nicely building on current knowledge. The manuscript may benefit by paying attention to the following aspects:
(1)
The authors introduce the topic of chromatin, and the DNA-centered methods, in a clear manner. However, the main aspect of this review - the MS-based methods and studies - should be more comprehensive, and explained in a more coherent way.
Locus-specificity issue: The authors have neglected the major limitation of MS-based methods to study chromatin - lacking locus-specificity. First, in section 4, they describe the study of histone modifications using MS. However, the authors should emphasize the fact that this method reports the total amount of a modification, on the entire genome, without any locus-information (this was monitored, for example, along the cell cycle by the Groth lab in 10.1101/gad.256354.114)
Most importantly, the authors should add a section to the review, where they address and discuss the relevant literature in this key aspect. For example, the authors can discuss works using short sequence baits ( 10.1101/gr.081711.108, 10.1186/s12864-015-2158-0 and many more). Additionally, this limitation has also been tackled by using a protein binding a specific DNA region to isolate a locus of interest (10.1016/j.cell.2008.11.045 & 10.1371/journal.pone.0026109 for example).
Furthermore, the authors can elaborate on studying proteins involved in chromatin in the nucleus versus in the cytoplasm. This is not mentioned, however can be very informative (e.g. histone pools in cytoplasm vs nucleus). Additionally, the authors did not cite landmark works and scientists in this field, which are key to the main topic of this review: The first study from 2003 using MS to study chromatin (10.1016/S1044-0305(03)00204-6), and identifying recruited factors to chromatin using MS
(10.1074/mcp.M110.005371–1 & 10.1073/pnas.1502971112). A beautiful paper by the Reinberg lab (10.1016/j.cell.2019.10.009), using a CRISPR-biotinylation system to tag specific loci should also be covered in this review.
MS Techniques: The authors explain the sequencing-based techniques in a clear and straight-forward manner. However, the mass spectrometry is in the center of this review, and the authors should better explain any MS-based methods, ideally with a figure/table that will help the readers compare. This will also emphasize the differences between these methods, which is crucial for this review. In line with this, part of section 3, starting from line 206, should explain the basics of the MS methods in a simplified manner that is comprehensible for readers who are not experts in MS. Alternatively, the authors should refer the reader to a review which explores these methods.
These clarifications should be carried out throughout the section, omitting some details of specific experimental conditions should be considered (concentrations for example), and jargon (e.g. “middle-down”) should be explained and elaborated on. What, for example, are the differences between “D6-acetic anhydride [61], to propionic anhydride [62] to NHS-propionate [63] to phenyl isocyanate” (line 259)?
(2) Referencing of published literature
The manuscript would benefit from better referencing, especially in the introductory sections (section 1 & 2). I suggest the authors examine this text carefully, add references to the claims, and make sure they cite papers relevant for their claims. I will list below a few examples of missing / need of refined references:
-No references are mentioned, but needed: Lines 55-60, Lines 220-224, Lines 261-264.
-The authors mention studying combinatorial histone codes but do not two studies that actually did this already with different strategies: (i) A technique to study combinatorially modified nucleosomes: 10.1126/science.aad7701 (ii) Combinatorial iChip: 10.1016/j.molcel.2016.07.023
-The authors should also mention or discuss using restriction enzymes to study chromatin (10.1007/978-1-61779-477-3_6). This was also recently harnessed for more contemporary platforms (10.1038/s41592-019-0730-2).
(3) Figures & Tables
- The review will benefit from, referencing or adding a table or a section discussing the types of proteins that bind DNA more broadly - with protein names, functions, etc.
- Line 181 Does this line have a typo? Is this correct? (notice dates): “Labs that use it was determined by the number of unique last authors from method-specific search results on PubMed between 03/24/2020 and 03/25/2020”. This repeats in the following line.
- Table 1: “chromatin state” is not an informative description. Please revise. For example “Methylated DNA”, “Exposed DNA”, as the methods do not probe the same thing. Additionally, the Methods difficulty is in a range between medium and high.I would suggest expanding this to a normal scale. For example, DNAse, Mnase,and ATAC can definitely be considered easy as compared to other methods. Lastly, I think it is crucial that the authors add another column to the table of whether the method requires fixation or not, as this is an important feature that has been debated in the field (10.7554/eLife.22280).
(4) Other comments:
Line 51 - The authors should introduce histone methylation as chromatin mark and not as an example through HP1.
Line 54 - The authors discuss how readers and transcription factors regulate chromatin accessibility, which may be confusing as “readers” usually refers to proteins that bind histones and histone marks. However, some canonical chromatin regulators bind DNA and then move nucleosomes (see for example 10.1038/379844a0).
Line 326 - change “he validated the method” to “the method was validated”.
Line 336 “are now a long list” - what is the estimate? It would be helpful to have a numerical estimate.
Line 368 “we have a long library”. Please revise.
